**Data Availability Statement:** All relevant data are within the manuscript.

**Funding:** The authors received no specific funding for this work.

# Pollen trapping as Honeybees pollination management and identification of dominant pollinators of Guizotia abyssinica (L.f)

**Kasim Roba Jilo** ✉️\*, **Gemmechis Legesse Yadeta, Tolera Kumsa Gemmeda**

Oromia Agricultural Research Institute, Holeta Bee Research Center, Holeta, Ethiopia

\* kasimroba7@gmail.com

## Abstract

Pollination is one of the most fascinating aspects of insect-plant interactions. Pollen is the male reproductive element of flowering plants, gathered by foraging Honeybees from the male parts of flowering plants called the anther. Guizotia abyssinica (L.f) is an important oil-seed crop cultivated in Ethiopia and India, which belongs to the family Asteraceae. Although self-incompatibility is found in Guizotia abyssinica, a higher seed set is experienced in places with an active Honeybees population. Agricultural practices usually focus on inputs such as fertilizer application to improve seed yield of Guizotia abyssinica. However, these practices have little effect on yield if the availability of insect pollination is too low. To fill this gap, an experiment was carried out at Dandi district, West Shao zone, Oromia, Ethiopia to see the effect of Honeybees pollination management as an agronomic input. Pollination management of Honeybees was tested under feeding and non-feeding of colonies management. Pollen was trapped with and without sugar syrup feeding. The results discovered that Honeybees fed sugar syrup collected much more pollen than colonies not fed sugar syrup. The proportion of Guizotia abyssinica pollen collected through sugar syrup feeding of the colony was greater (62.2%), compared to the proportion of Guizotia abyssinica pollen trapped without sugar syrup feeding (37.8%). This indicates that sugar syrup feeding enhances the collection of pollen and probably enhances the pollination efficiency of Honeybees since they visit frequently to fulfill their daily protein requirement of pollen. Therefore, Honeybees pollination services should be included as one of the agronomic inputs with sugar syrup feeding as pollination management that might increase the yield of Guizotia abyssinica since it increases visiting frequency.

## Introduction

Pollinators are a functional group that ensures cross-pollination in wild plant populations and yields in major crops [1]. Pollination plays a significant role in the agriculture sector and serves as a basic pillar for crop production [2]. Insect-plant interactions fascinate researchers, especially the process of pollination [3]. Insects provide ecosystem services that are beneficial for human life and other living things [4]. Bees, especially, are important in bringing about the

**Competing interests:** The authors have declared that no competing interests exist.

pollination of many plant species. In temperate regions, attention has been given to the deliberate manipulation of these pollen vectors to increase crop yields [5]. Nectar sugar composition has often been related to the pollination syndrome of the plant species [6]. The principal insect species used for pollination of crops is Apis mellifera [7]. Apis mellifera is suitable for the purposes of pollen vector manipulation [5]. Pollination is the movement of pollen from the anthers of a flower to the stigma of the same or a different flower [8].

From an applied perspective, species richness is the ultimate "score card" in efforts to preserve biodiversity in the face of increasing environmental pressures and climate change resulting from human activity [9]. It is common practice among ecologists to complete the description of a community by one or two numbers expressing the diversity or the evenness of the community [10]. Numerous evolutionary lineages in the early divergent angiosperm possess flowers with a distinctive pollinator trapping mechanism, in which floral phenological events are very precisely timed in relation with pollinator activity patterns [11].

Species richness, evenness, and biodiversity are important concepts in the study of species diversity [12]. There are several methods to measure these concepts, such as the Simpson, Camargo, and Smith & Wilson indices for evenness and the Shannon-Wiener, Brillouin, and Simpson indices for biodiversity. It is important to know the suitable index and its measurement method to study species diversity. *G. abyssinica* (L.f) Cass. is an important oilseed crop cultivated in Ethiopia and India, belonging to the family Asteraceae, tribe Helianthus, and sub-tribe Coreopsinidae and grown over an area of 232.1 thousand hectares with an average production of 76.2 thousand tons in India [13]. Niger seeds are utilized for human consumption, as bird feed and extraction of oil [14]. The genus Guizotia belongs to the family of Compositae, tribe Heliantheae, subtribe Coreopsidinae [15].

Pollinator diversity and abundance have declined globally, raising concerns about a pollination crisis of crops and wild plants [16]. To ensure the security of our pollinator-dependent crop species, it is imperative to characterize the mechanisms and practices that can enhance pollinator ecosystem services in managed landscapes [17]. The populations of Honeybees (Apis mellifera) and non-Apis bees in the United States have grown increasingly important, as declines in their populations have the potential to impact food security due to loss of pollination services [18]. For these reasons, pollination management is important to overcome the problems of yield loss obtained from pollination services. Our research was conducted on pollination management of Honeybees on G. abyssinica (Lf). to address this issue.

A simple count of the number of species in a sample is usually a biased underestimate of the true number of species because increasing the sampling effort inevitably increases the number of species observed [9]. In nature, only 5% of crops are self-pollinated, while the remaining 95% are cross-pollinated, with 10% depending on wind and 85% on animal pollination [19]. Insect pollination alone accounts for 90% of animal pollination. Species richness is a diversity of order 0, which means it is completely insensitive to species abundances and many species diversity indices can be converted by an algebraic transformation to Hill numbers [9].

Pollination is a vital process in the reproduction of flowering plants, which leads to fertilization and fruit/seed setting, and is one of the most important mechanisms in the maintenance and conservation of biodiversity and benefits society by increasing food security and improving livelihoods [20]. Worldwide, 90 percent food supply is contributed by 82 commodities assigned to plant species and bees are pollinators of 63 (i.e.70%) of these plant species and are the most important known pollinators of 39 (48%) of these plant species [20]. Bees are keystone organisms that sustain human life on earth through their pollination services. However, very little is known about functional groups and indicator species of bee communities from agricultural landscapes in Sub-Saharan Africa [21]. Pollination is one of the most fascinating aspects of insect-plant interactions [3].

## Materials and methods

The study was conducted in Dandi district at Grinch, West Shao, Oromia regional state, Ethiopia in 2020–2022. Colonies were established for pollination management of G. abyssinica using Honeybees, which were fed and none fed sugar syrup as an agronomic input. The flower visitors were collected with fine mesh sweep nets and got identified following [22]. Species evenness (or relative species abundance) in a community is another factor that affects diversity. One of the most frequent methods of evenness measurement is Simpson. Simpson's evenness index [12].

### Pinning collected pollinators of *G. abyssinica*

Diversity of species collected and added to alcohol for preservation and brought from the experimental area, mounted, pinned (Fig 1) for identification and then taking the dried specimens and placing them on a foam pad such that they are either upside down or on their sides were used following [23].

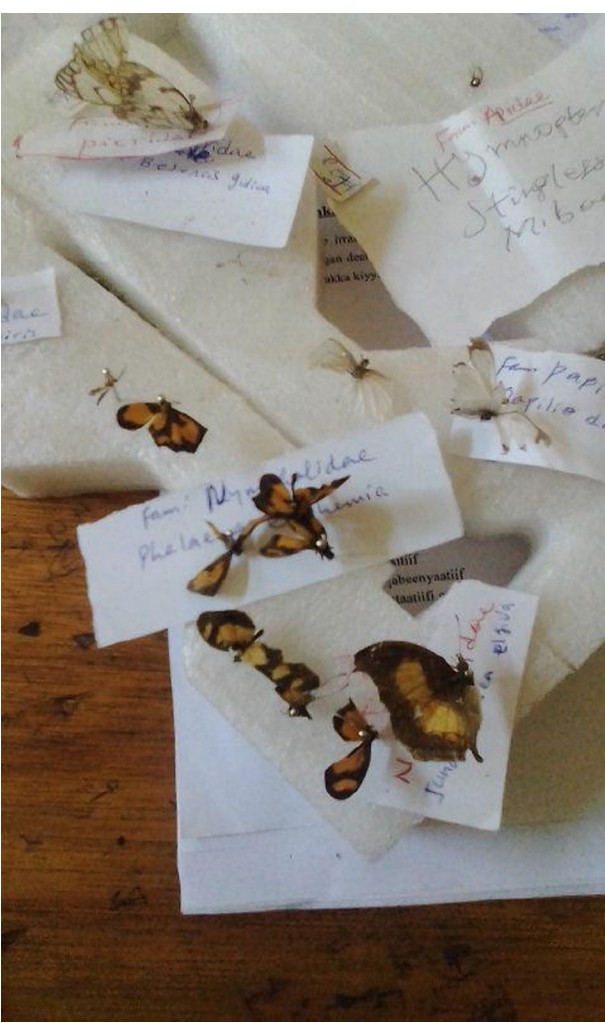
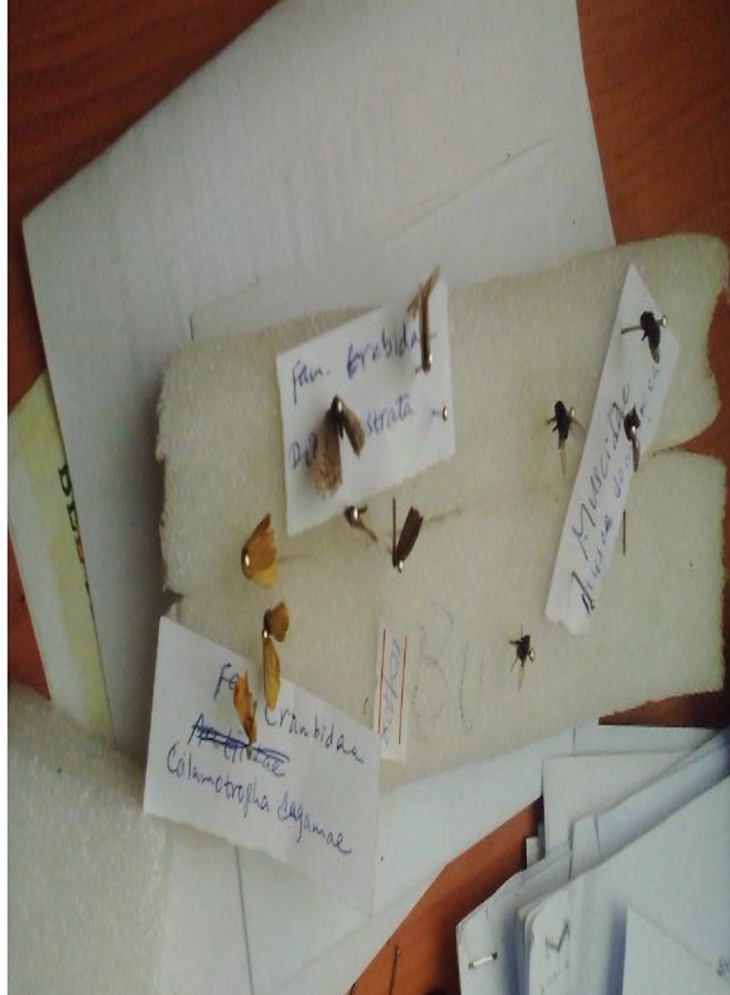

**Fig 1. Mounting and labeling of insect's specimen for identification.**

### Bee management

Special management was given to colonies used for pollen trapping experiments to ensure sufficient pollen by feeding and none feeding to study the difference to use Honeybees as agronomic inputs for pollination management. The experiment was conducted on active and strong colonies with equal resources, such as brood and colony strength, from the same area during the active period of September-October when pollen availability was high [24].

### Bee feeding

Sugar syrup ratio of 1:1 was used to stimulate them to collect more pollen. Pollen was collected until Niger stopped pollen production and shaded. Sugar syrup (1:1) was given to each colony by internal feeding methods at Gunch until the experiment was completed at every 48 hrs. following methods used by [25].

### Flower visitors

Pollinators abundance was recorded when floral density found in the field (Fig 2). The experiment was divided into eight plots for sampling pollinators from each plot in 10 minutes each when it was bloomed 50% -75%. Sampling was done the whole day continuosly until all pollinators were absent from the flowers of G.abyssinica. Samplings were done through visual counting and trapping of flower visitors for 10 minutes per predetermined transect of $1m^2$ area, and this was repeated five times by 10 mintues gap each between two subsequent transects following [3].

Honeybees' colonies were established at Grinch, Dandi district, west Shao, Oromia regional state, Ethiopia. Pollen trap was fitted at the entrance of bee hives having the pollen trapping efficiency of 16% and pollen loads was collected daily until it stops flowering or shaded. The pollen samples were placed in a clean paper bag and left for 24 hrs. to dry at room temperature. The pollen grains were collected and sorted into colors and identified to the genus and species level.

### Maintaining the specimens

All species were identified as the same species or group grouped in rows or sections. A new row or section is started for each new taxa which, makes it easy to see the groupings. It is easiest to have a determination label precede the row or section of specimens following methods used by [23].

### Pollen trapping

Worker bees collect pollen from flowers and carry it back to the hive packed in pellets on the pollen baskets on the rear legs (Fig 3). By encouraging returning field bees to enter the hive through small holes in a punched tyce, the pellets of pollen can be scraped from the legs and collected in a suitable tray and Pollen collected every two days following methods used by [24]. Traps are fitted and worked continuously until the experiment ends for three weeks following methods used by [4].

Worker bees collect pollen from flowers and carry it back to the hive packed in pellets on the pollen baskets on the rear legs. Hives placed at the experimental area attaching pollen traps on its entry for pollen trapping Unsorted and sorted pollen trapped from hives collected by Honeybees and sorted image of *G.abyssinica (*L.f) pollen (Fig 4).

By encouraging returning field bees to enter the hive through small holes in a punched tyce, the pellets of pollen can be scraped from the legs and collected in a suitable tray and Pollen

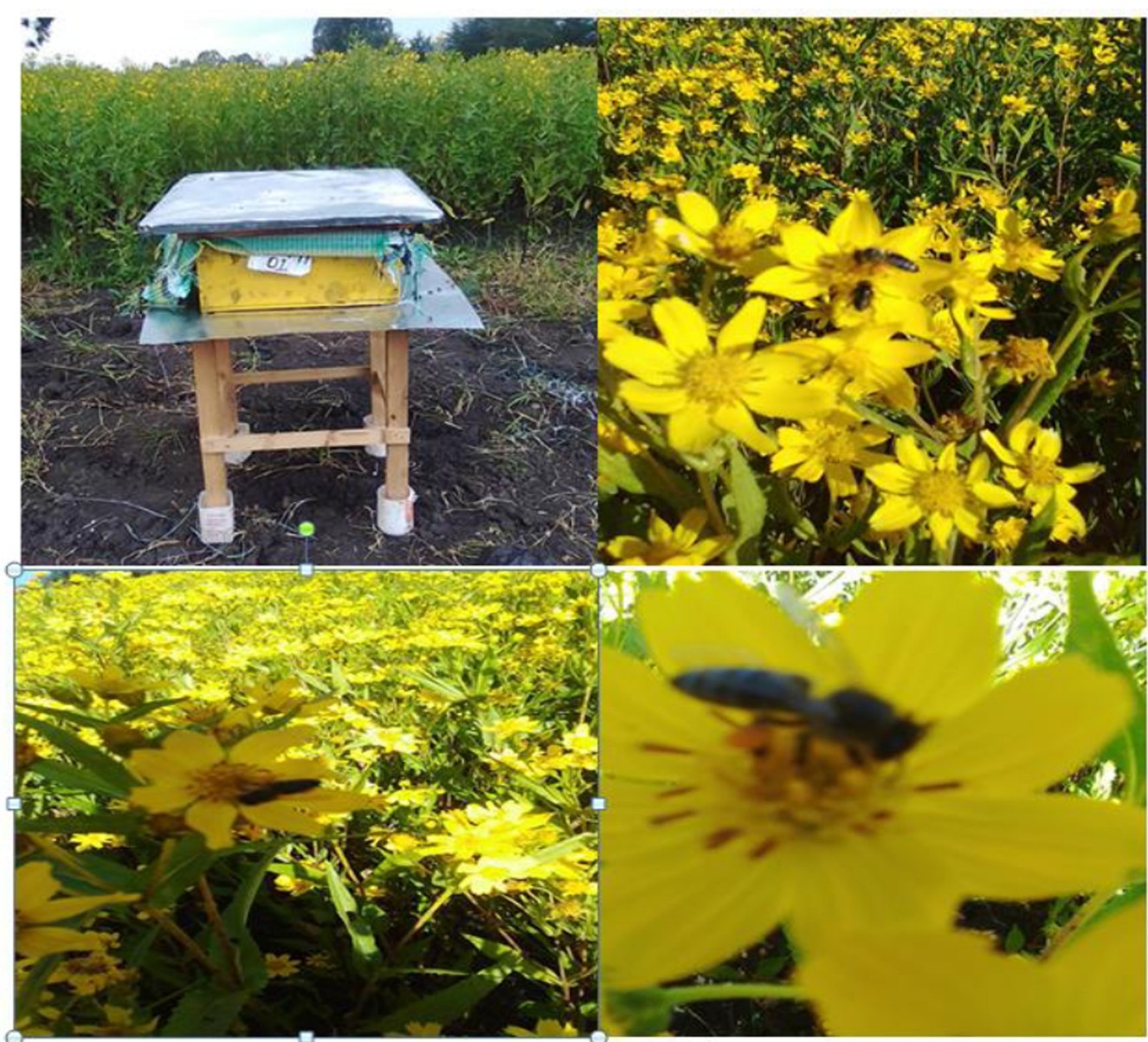

**Fig 2. Colony established for pollination purpose of Guizotia abyssinica (L. f).**

collected every two days following methods used by [24]. Traps are fitted and worked continuously until the experiment ends for three weeks following methods used by [4].

## Data analysis

Descriptive statistics, Anova and Shannon-Wiener Index (H') (Shannon & Weaver, 1949); were used to estimate pollinators diversity in each plot. Statistical analyses were conducted to determine the importance of some taxa relative to others. To explore whether there were statistically significant differences in occurrences or proportional abundance / species richness between different taxonomic groups (genera, families, etc), chi-square tests were applied [21].

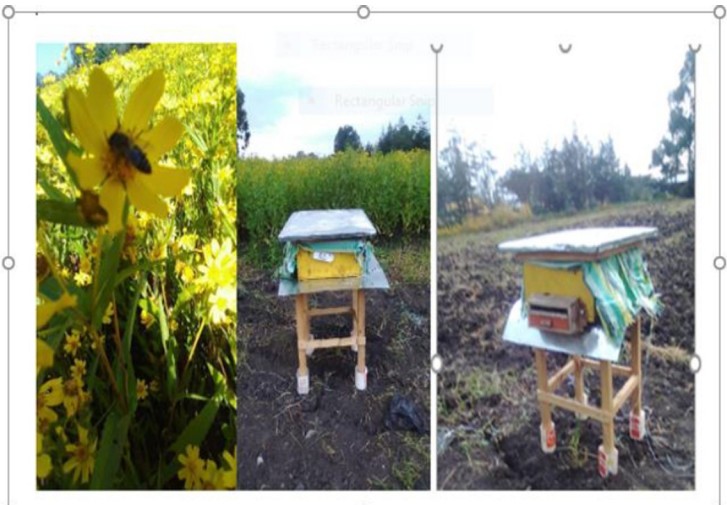

**Fig 3. Hives placed at the experimental area attaching pollen traps on its entry for pollen trapping.**

## Results and discussions

The results of the experiment showed that Honeybees collected much more pollen daily when colonies were fed sugar syrup than when they were not fed sugar syrup. The difference was statistically significant at p-value of 0.001, as shown in (Fig 5). A total of 196.34 grams of pollen

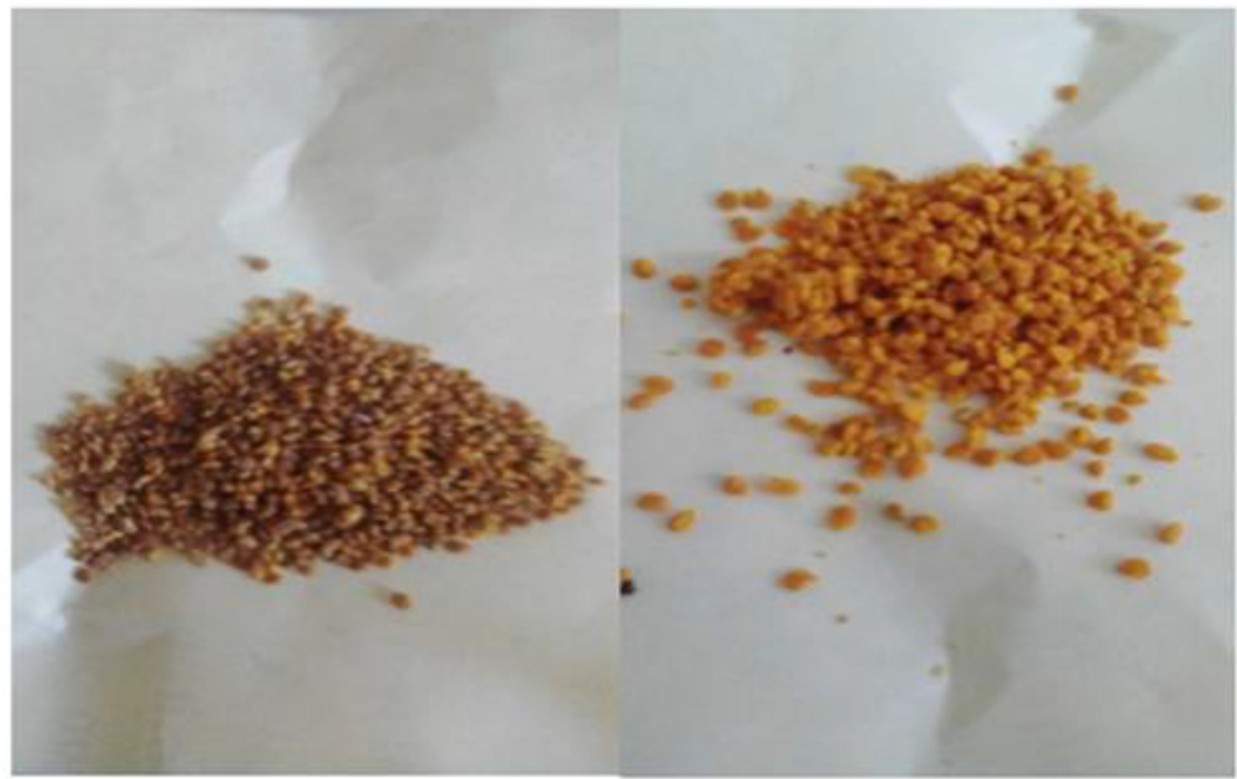

**Fig 4. Unsorted and sorted pollen trapped from hives collected by Honeybees and sorted image of G. abyssinica (L.f) pollen.**

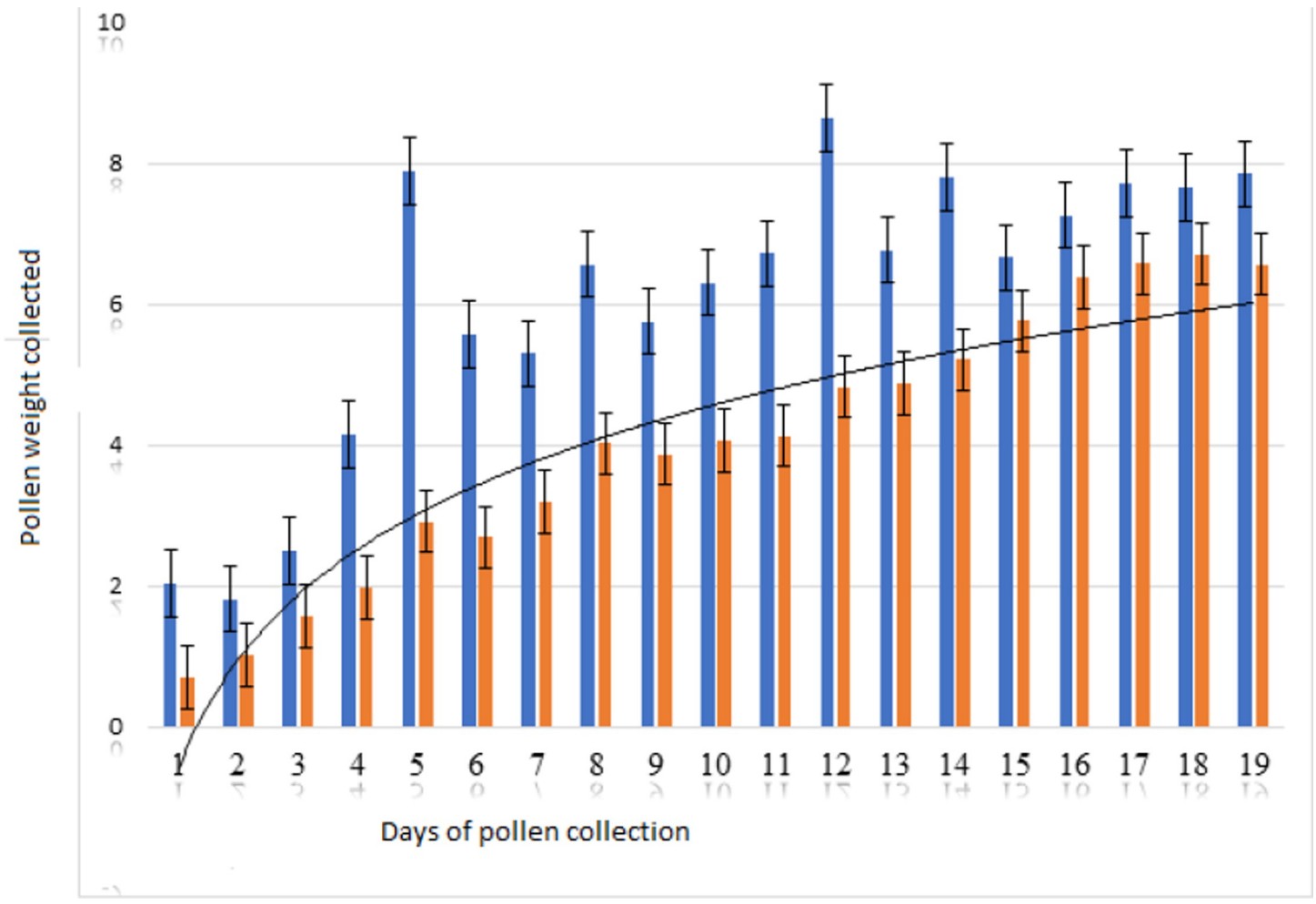

**Fig 5. Weight of pollen collected by pollen trap while pollinating G. abyssinica.**

was trapped by the pollen trap as part of the pollination management strategy to enhance bee visiting frequency and increase pollination. The pollen samples were sorted into different colors, identified to the species level, and classified under eight species [26]. This implies that Honeybees collect more pollen when pollen is trapped to fulfill their daily requirement of pollen, which probably increases pollination of G. abyssinica (L.f) The proportion of G. abyssinica pollen collected through sugar syrup feeding of the colony for three weeks until the experiment ended was greater (62.2%) compared to the proportion of pollen trapped without colony feeding with sugar syrup (37.8%). This indicates that sugar syrup feeding enhances the collection of Guizotia abyssinica (L.f) pollen and probably the pollination efficiency of Honeybees. Pollen weight was inferred from the pollen loads collected through pollen traps following [27]. Therefore, Honeybees pollination services should be included as one of the agronomic inputs to meet the yield of Guizotia abyssinica (L.f) through pollination management of Honeybees.

Color name has its own color image as shown in Fig 6 below. Eight species of pollen collected by Honeybees were sorted and identified for both colonies feed sugar syrup and none feed sugar syrup as shown in Table 1 below. Fed colonies collected 68.8% of Acacia abyssinica, whereas none fed colonies collected 31.3% of A. abyssinica which has Brow Mod color identified based genetics color bar code manuals. 54.7% of G. abyssinica was collected by fed

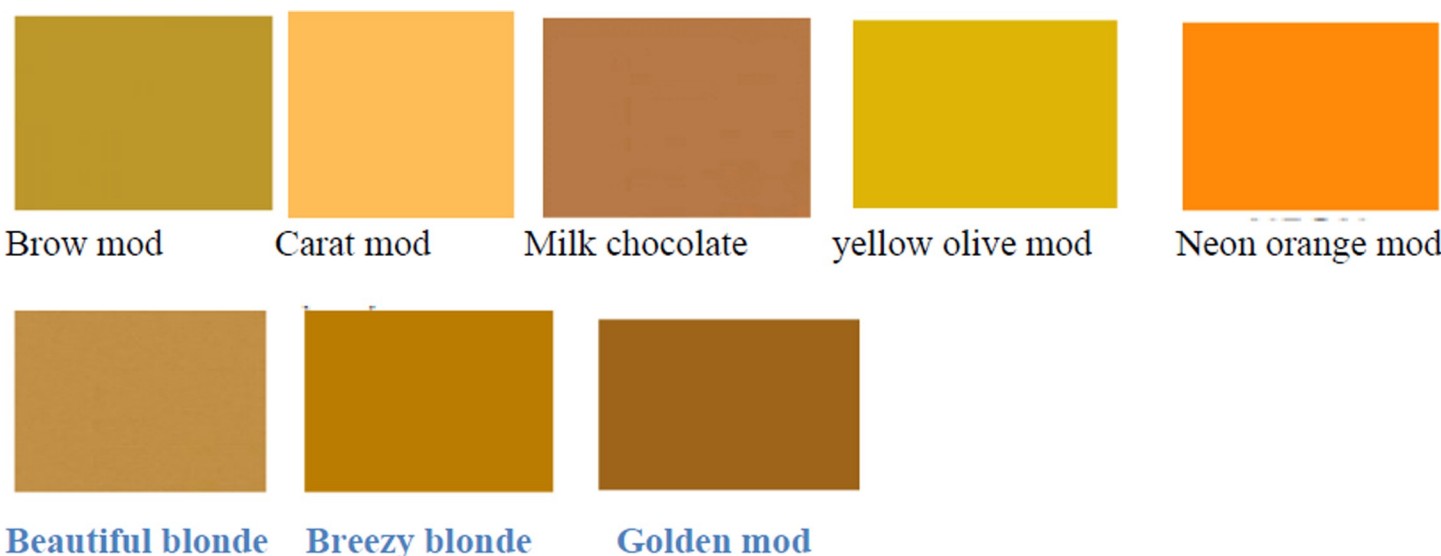

**Fig 6. Colors of pollen identified after sorted into different colors according to their species.**

colonies where as 45.3% collected by none fed colonies which has Neon orange Mod color. Zantedeschia aethiopica, Vicia faba and Sesamum indicum pollen were only collected by fed colines which might be due to colonies strength enabled collection from far areas where as Clausena anisate was only collected by none fed colonies in small amount which may be due to weakness of the colonies or their preference.

18 species of pollinators of G. abyssinica were recorded and identified which are classified under 12 familes, 83.2% were Apis mellifera where as 8.49% were Musca.domestica (Table 2).

Honeybees visited flowers of G. abyssinica (L.f) for nectar and pollen as closely observed visually which is similar to results reported by [22]. Around 83.2% pollinators were belonging to family Apidae which were Apis mellifera mainly in morning as well as in afternoon than all pollinators; this is in line with the reports of [21]. Foraging speed in terms of time spent on

**Table 1. Pollen trapped by Honeybees, its color and color name, family and species identified.**

| Species of pollen collected | Families of species | Frequency and their percentage | | Colors' names |
|---|---|---|---|---|
| | | Not fed | fed | |
| Acacia abyssinica | Fabaceae | 10<br>31.3% | 22<br>68.8% | Brow Mod |
| Acanthus sennii | Acanthaceae | 2<br>40% | 3<br>60% | Carat Mod |
| Clausena anisata | Rutaceae | 1<br>100% | 0<br>0% | Milk Chocolate |
| Eucalyptus camaldulensis | Myrtaceae | 13<br>30.95% | 29<br>69.1% | Yellow olive Mod |
| Guizotia abyssinica | Asteraceae | 29<br>45.3% | 35<br>54.7% | Neon orange Mod |
| Sesamum indicum | Pedaliaceae | 0<br>0% | 1<br>100% | Beautiful Blonde |
| Vicia faba | Fabaceae | 0<br>0% | 1<br>100% | Breezy Blonde |
| Zantedeschia aethiopica | Araceae | 0<br>0% | 11<br>100% | Golden Honey |

**Table 2. Scientific name of pollinators, their relative abundance and total species occurred at time classified as below.**

| Scientific name of pollinators | Families of pollinators | Relative abundance of each species *at a given time below* | | | | | | Total % |
|---|---|---|---|---|---|---|---|---|
| | | 10:20–11:50AM | 1:15–2:45PM | 2: 45–4:15 PM | 4:15–5:45PM | 7:20–8:50AM | 8:50–10:20AM | |
| *Amata alicia* | Erebidae | 0.12 | 0 | 0 | 0 | 0 | 0 | 0.12 |
| *Anthene princeps* | Lycaenidae | 0.06 | 0 | 0 | 0 | 0 | 0 | 0.06 |
| *Apis mellifera* | Apidae | 20.7 | 17.91 | 12.61 | 2.47 | 8.72 | 20.7 | 83.20 |
| *Calliphora vicina* | Calliphoridae | 0 | 0 | 1.00 | 0.18 | 0 | 0 | 1.18 |
| *Cosmetra tumulata* | Tortricidae | 0.06 | 0 | 0 | 0 | 0 | 0 | 0.06 |
| *Epiphora fournierae* | Saturniidae | 0 | 0 | 0 | 0 | 0.06 | 0 | 0.06 |
| *Eupeodes luniger* | Syrphidae | 0.35 | 0.41 | 0.29 | 0 | 0.41 | 0.41 | 1.89 |
| *Graphomya maculata* | Muscidae | 0.06 | 0 | 0 | 0 | 0 | 0 | 0.06 |
| *Lexias pardalis* | Nymphalidae | 0.06 | 0.06 | 0 | 0 | 0 | 0 | 0.12 |
| *Meliscaeva auricollis* | Syrphidae | 0.59 | 0.18 | 0 | 0 | 0 | 0.47 | 1.24 |
| *Musca.domestica* | Muscidae | 1.06 | 1.36 | 1.00 | 0.53 | 2.89 | 1.65 | 8.49 |
| *Myathropa florea* | Syrphidae | 0.18 | 0.29 | 0.24 | 0 | 0.82 | 0.82 | 2.36 |
| *Phalaena euphemia* | Nymphalidae | 0 | 0 | 0 | 0 | 0 | 0.12 | 0.12 |
| *Platycorynus dejeani* | Chrysomelidae | 0 | 0 | 0.06 | 0 | 0 | 0 | 0.06 |
| *Syrphus ribesii* | Syrphidae | 0 | 0 | 0.71 | 0 | 0 | 0 | 0.71 |
| *Vespula vulgaris* | Vespidae | 0.06 | 0 | 0 | 0 | 0 | 0 | 0.06 |
| *Xylocopa longespinosa* | Apidae | 0 | 0 | 0.06 | 0 | 0 | 0 | 0.06 |
| *papilio dardanus* | Papilionidae | 0.06 | 0.12 | 0 | 0 | 0 | 0 | 0.18 |

each flower varied between bee species [28]. Pollinators other than Honeybees are also extremely valuable although their value is difficult to estimate [29]. Time has significant effect on species distributions with p value of < 0.001. A total of 18 insect species representing six families visited Niger flowers. These species and families mentioned in Table 2 were visitors of Niger in our experiment

Time has signicant effects on these distribution of pollinators species and their families at P<0.001. Peak bee activity was recorded between 10:20–11: 50 AM**;** Similarly, to our findings a study conducted by [22]. showed that peak bee's activity was recorded between 11.00 and 13.00 hrs. and this has the highest species richness which was 16 (Table 3). Which will increase primary productivity and this implies there is great pollinators diversity for pollination purpose which will play a key role in crop yield production and productivity for this area and which is key results for pollinators diversity conservation. Whereas the lowest species richness was 8 recorded between 8:50–10:20AM. 1.74. The highest species diversity was recorded between 2: 45–3:15 PM whereas the highest evenness of 0.68 was recorded between 2: 45–3:15 PM.

The highest species richness, diversity, and evenness were observed in the morning at the plots mentioned below (Table 4). In the first cycle, more species richness was observed at plots

**Table 3. Pollinators diversity, richness and evenness at a given exposure time.**

| Time | Diversity | Richness | Evenness |
|---|---|---|---|
| 1:15–2:45 PM | 1.00 | 12 | 0.40 |
| 10:20–11: 50 AM | 1.25 | 16 | 0.45 |
| 2: 45–3:15 PM | 1.74 | 13 | 0.68 |
| 4:15–5:45 PM | 0.64 | 13 | 0.25 |
| 7:20–8:50 AM | 0.85 | 14 | 0.32 |
| 8:50–10:20 AM | 0.62 | 8 | 0.30 |

**Table 4. Plots, pollinators diversity, species richness and evenness.**

| plots | H | Richness | Evenness |
|---|---|---|---|
| **1st cycle** | | | |
| A1 | 1.21 | 4 | 0.88 |
| B1 | 1.04 | 3 | 0.95 |
| C1 | 0.56 | 3 | 0.81 |
| D1 | 1.24 | 4 | 0.89 |
| E1 | 0.85 | 4 | 0.77 |
| F1 | 0.70 | 3 | 0.64 |
| G1 | 0.76 | 3 | 0.69 |
| H1 | 0.42 | 3 | 0.38 |
| I1 | 0.75 | 4 | 0.54 |
| **2nd cycle** | | | |
| A2 | 0.79 | 4 | 0.57 |
| B2 | 0.81 | 4 | 0.59 |
| C2 | 0.78 | 3 | 0.71 |
| D2 | 0.83 | 6 | 0.46 |
| E2 | 0.20 | 3 | 0.18 |
| F2 | 0.51 | 4 | 0.37 |
| G2 | 0.10 | 2 | 0.14 |
| H2 | 0.39 | 3 | 0.35 |
| I2 | 0.32 | 3 | 0.29 |
| **3rd cycle** | | | |
| A3 | 0.63 | 5 | 0.39 |
| B3 | 0.34 | 4 | 0.25 |
| C3 | 0.48 | 4 | 0.35 |
| D3 | 0.60 | 4 | 0.43 |
| E3 | 0.95 | 4 | 0.86 |
| F3 | 0.43 | 3 | 0.39 |
| G3 | 0.31 | 4 | 0.23 |
| H3 | 0.46 | 4 | 0.33 |
| I3 | 0.29 | 2 | 0.41 |
| **4th cycle** | | | |
| A4 | 0.00 | 1 | Nan |
| B4 | 0.13 | 2 | 0.19 |
| C4 | 0.33 | 4 | 0.24 |
| D4 | 0.40 | 3 | 0.36 |
| E4 | 0.72 | 4 | 0.52 |
| F4 | 0.73 | 4 | 0.53 |
| G4 | 0.55 | 4 | 0.39 |
| H4 | 0.50 | 3 | 0.46 |
| I4 | 0.67 | 4 | 0.48 |
| **5th cycle** | | | |
| A5 | 0.00 | 1 | Nan |
| B5 | 0.00 | 1 | Nan |
| C5 | 0.99 | 5 | 0.61 |
| D5 | 0.54 | 5 | 0.34 |
| E5 | 0.77 | 5 | 0.48 |
| F5 | 0.52 | 2 | 0.75 |

(*Continued*)

**Table 4.** (Continued)

| 1<sup>st</sup> cycle | | | |
|---|---|---|---|

| plots | H | Richness | Evenness |
|---|---|---|---|
| G5 | 0.91 | 3 | 0.83 |
| H5 | 0.81 | 3 | 0.74 |
| I5 | 0.51 | 2 | 0.74 |
| 6<sup>th</sup> cycle | | | |
| A6 | 0.00 | 1 | Nan |
| B6 | 0.00 | 1 | Nan |
| C6 | 0.56 | 2 | 0.81 |
| D6 | 0.00 | 1 | Nan |
| E6 | 0.66 | 2 | 0.95 |
| F6 | 0.56 | 2 | 0.81 |
| G6 | 0.00 | 0 | 0.00 |
| H6 | 0.00 | 0 | 0.00 |
| I6 | 0.00 | 0 | 0.00 |

Letters from A1 up to I1 represents plots 1m*1m in the morning while Letters from A2 up not 12 resents plots size of 1m*1m taken in the afternoon from where insects were trapped by an interval of 10 minutes both in the morning and afternoon for identification.

A1, D1, E1, and I1 than the others. High diversity was observed at plots D1, A1, and B1, respectively, than the others, whereas the highest evenness was observed at plots B1, D1, and A1, respectively, than the others. The highest species diversity was observed at plots E3, A3, and D3, respectively, whereas the highest species evenness was observed at plot E3, more than all the others. The highest species diversity was observed at plots C5, G5, H5, E5, D5, and F5, respectively, whereas the highest species evenness was observed at plots G5, H5, I5, F5, and C5, respectively, more than others. In the sixth cycle, the highest species richness was observed at plots C6, E6, and F6, respectively. The highest diversity was observed at plots E6, C6, and F6, respectively, whereas the evenness was observed at plots E6, C6, and F6, respectively, more than all the others.

## Conclusion

In conclusion, the results of the study indicated that Honeybees fed with sugar syrup collected much more pollen than colonies not fed with sugar syrup. The proportion of Guizotia abyssinica (L.f) pollen collected through sugar syrup feeding of the colony was greater (62.2%), compared to the proportion of G.abyssica (L.f) pollen trapped without colony feeding with sugar syrup (37.8%). This indicates that sugar syrup feeding enhances the collection of Guizotia abyssinica (L.f) pollen and probably the pollination efficiency of Honeybees. Therefore, Honeybees pollination services should be included as one of the agronomic inputs that might increase the yield since it visits frequently and may pollinate it, increasing the yield of Guizotia abyssinica (L.f) through pollination management of Honeybees. 83.20% of pollinators of Guizotia abyssinica (L.f) were Honeybees. This is the first novel work using sugar syrup as pollination management of Honeybees for yield increment rather than using only fertilizer. The highest species richness, diversity, and evenness were seen at plots in the morning than in the afternoon. Knowing the status of all species and their pollination role for all crops that need pollination is recommended for further research.

## Acknowledgments

First and foremost, I would like to express my gratitude to the Ethiopian Institute of Agricultural Research. I would also like to acknowledge Dr. Zemzem Ahmed, for her inspiration while I was writing the paper and collecting data from a long distance.

## Author Contributions

**Conceptualization:** Kasim Roba Jilo.

**Data curation:** Kasim Roba Jilo.

**Formal analysis:** Kasim Roba Jilo.

**Investigation:** Kasim Roba Jilo.

**Methodology:** Kasim Roba Jilo, Gemmechis Legesse Yadeta, Tolera Kumsa Gemmeda.

**Validation:** Kasim Roba Jilo, Gemmechis Legesse Yadeta.

**Visualization:** Kasim Roba Jilo.

**Writing – original draft:** Kasim Roba Jilo.

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
