## [Decision Letter · Decision Letter 0]

8 Dec 2023

PONE-D-23-28714Pollen Trapping As Honeybees Pollination Management and Identification of a Dominant Pollinators of Guizotia abyssinica L.PLOS ONE

Dear Dr. Jilo,

Thank you for submitting your manuscript to PLOS ONE. After careful consideration, we feel that it has merit but does not fully meet PLOS ONE’s publication criteria as it currently stands. Therefore, we invite you to submit a revised version of the manuscript that addresses the points raised during the review process.

Dear

Currently we got the reviewers comments on your manuscript related to pollination of G. abyssinica. Please be sure to address all comments point by point for both reviewers and follow the PlosOne instructions.

Regards

We look forward to receiving your revised manuscript.

Kind regards,

Rachid Bouharroud

Academic Editor

PLOS ONE

Journal Requirements:

4. We suggest you thoroughly copyedit your manuscript for language usage, spelling, and grammar. If you do not know anyone who can help you do this, you may wish to consider employing a professional scientific editing service.  

'funded for data collection only". 

6. Thank you for stating the following in your Competing Interests section:  

"26761".

7. In your Data Availability statement, you have not specified where the minimal data set underlying the results described in your manuscript can be found. PLOS defines a study's minimal data set as the underlying data used to reach the conclusions drawn in the manuscript and any additional data required to replicate the reported study findings in their entirety. All PLOS journals require that the minimal data set be made fully available. For more information about our data policy, please see http://journals.plos.org/plosone/s/data-availability.

8. We note that you have stated that you will provide repository information for your data at acceptance. Should your manuscript be accepted for publication, we will hold it until you provide the relevant accession numbers or DOIs necessary to access your data. If you wish to make changes to your Data Availability statement, please describe these changes in your cover letter and we will update your Data Availability statement to reflect the information you provide.

9. Please amend your list of authors on the manuscript to ensure that each author is linked to an affiliation. Authors’ affiliations should reflect the institution where the work was done (if authors moved subsequently, you can also list the new affiliation stating “current affiliation:….” as necessary).

Reviewers' comments:

Reviewer's Responses to Questions

**Comments to the Author**

1. Is the manuscript technically sound, and do the data support the conclusions?

Reviewer #1: Yes

Reviewer #2: No

2. Has the statistical analysis been performed appropriately and rigorously? 

Reviewer #1: Yes

Reviewer #2: No

3. Have the authors made all data underlying the findings in their manuscript fully available?

Reviewer #1: Yes

Reviewer #2: Yes

4. Is the manuscript presented in an intelligible fashion and written in standard English?

Reviewer #1: Yes

Reviewer #2: No

5. Review Comments to the Author

Reviewer #1: The manuscriptis technically fitted the journals scope and its conclusion is inline with a data. Please make a uniform how to write a citation in the body and reference throughout the manuscript consistently.

Reviewer #2: Reviewer Comments.

1. The scientific name of “Guizotia abyssinica” is not written in right form as French naturalists studied the plant and Verbesina oleifera was the first name given to niger. Polymnia abyssinica L was the first botanical description of the niger (Cassini, 1821). In 1905 following the Vienna Botanical Congress, the name Guizotia was conserved, and in 1930 at the Cambridge Botanical Congress the name Guizotia abyssinica (L. f.) Cass. was proposed as the correct name (Getinet and Sharma, 1996).

2. The introduction part is too long and ideas are repeated and don’t keep the flow of ideas.

3. The sub-topic species richness estimation in the introduction part is extra so, make the idea blend with the introduction and keep the flow of the idea.

4. Most Scientific names are not italicized.

5. Starting from the abstract the manuscript have gap problem, citation problem, ideas are repeated, Spelling error, and some sentence or paragraphs are not clear and all are highlighted in green colour.

6. Figure 1. Mounting and labeling of the insect’s specimen for identification didn’t seem the picture of honeybee and it’s not a clear image.

7. This manuscript explains the honeybee as a pollinator but in the result and discussion part the author explains other pollinators and I highlighted it in green.

8. Poor sentence construction in result and discussion.

9. Most citations are not correctly cited. E.g Steffan-dewenter and Westphal 2008 cited in the introduction are three authors in the reference section which can be written (Steffan et al., 2008).

10. The manuscript is not submitted based on PLOS ONE author’s guidelines it should include page no, continuous line number, Vancouver reference style, figure and table shouldn’t be found in the main manuscript file.

6. PLOS authors have the option to publish the peer review history of their article (what does this mean?). If published, this will include your full peer review and any attached files.

Reviewer #1: No

Reviewer #2: No

---

## [Author Response · Author response to Decision Letter 0]

26 Feb 2024

All response to reviewers uploaded as a rebuttal letter. please I have answered all comments given sorry for I didn't high light them. I hope you will consider it for publication. All things done by me I don't know what is being asked please go for publication it is original paper.

---

## [Editor Report · Decision Letter 1]

29 Feb 2024

PONE-D-23-28714R1Pollen Trapping As Honeybees Pollination Management and Identification of a Dominant Pollinators of Guizotia abyssinica L.PLOS ONE

Dear Dr. Jilo,

Thank you for submitting your manuscript to PLOS ONE. After careful consideration, we feel that it has merit but does not fully meet PLOS ONE’s publication criteria as it currently stands. Therefore, we invite you to submit a revised version of the manuscript that addresses the points raised during the review process.

**Dear Authors****This is to let you know that you didn't address all comments rised by reviewers. Please be more accurate and address all comments one by one and do not forgot to check comments in attached file. Do not use anymore such answers "I am a botanist"; "I am a researcher", only your work has a value in scientific sound manner.****Good luck**

We look forward to receiving your revised manuscript.

Kind regards,

Rachid Bouharroud

Academic Editor

PLOS ONE

---

## [Author Response · Author response to Decision Letter 1]

1 Mar 2024

I have sent response to reviewers here I need your help I have corrected all comments in text please for further I need your help if I missed any excuse.

---

## [Editor Report · Decision Letter 2]

12 Mar 2024

Pollen Trapping As Honeybees Pollination Management and Identification of a Dominant Pollinators of Guizotia abyssinica L.

PONE-D-23-28714R2

Dear Dr. Jilo,

We’re pleased to inform you that your manuscript has been judged scientifically suitable for publication and will be formally accepted for publication once it meets all outstanding technical requirements.

Kind regards,

Rachid Bouharroud

Academic Editor

PLOS ONE
---

## [Editor Report · Acceptance letter]

14 Aug 2024

PONE-D-23-28714R2 

PLOS ONE

Dear Dr. Jilo, 

I'm pleased to inform you that your manuscript has been deemed suitable for publication in PLOS ONE. Congratulations! Your manuscript is now being handed over to our production team.

Kind regards, 

on behalf of

Dr. Rachid Bouharroud 

Academic Editor

PLOS ONE